# The Efficacy of Semantics-Preserving Transformations in Self-Supervised Learning for Medical Ultrasound

**DOI:** 10.3390/bioengineering12080855

**Published:** 2025-08-08

**Authors:** Blake VanBerlo, Jesse Hoey, Alexander Wong, Robert Arntfield

**Affiliations:** 1David R. Cheriton School of Computer Science, University of Waterloo, Waterloo, ON N2L 3G1, Canada; 2Systems Design Engineering, University of Waterloo, Waterloo, ON N2L 3G1, Canada; 3Schulich School of Medicine and Dentistry, Western University, London, ON N6A 3K7, Canada

**Keywords:** data augmentation, machine learning, self-supervised learning, transfer learning, ultrasound

## Abstract

Data augmentation is a central component of joint embedding self-supervised learning (SSL). Approaches that work for natural images may not always be effective in medical imaging tasks. This study systematically investigated the impact of data augmentation and preprocessing strategies in SSL for lung ultrasound. Three data augmentation pipelines were assessed: (1) a baseline pipeline commonly used across imaging domains, (2) a novel semantic-preserving pipeline designed for ultrasound, and (3) a distilled set of the most effective transformations from both pipelines. Pretrained models were evaluated on multiple classification tasks: B-line detection, pleural effusion detection, and COVID-19 classification. Experiments revealed that semantics-preserving data augmentation resulted in the greatest performance for COVID-19 classification—a diagnostic task requiring global image context. Cropping-based methods yielded the greatest performance on the B-line and pleural effusion object classification tasks, which require strong local pattern recognition. Lastly, semantics-preserving ultrasound image preprocessing resulted in increased downstream performance for multiple tasks. Guidance regarding data augmentation and preprocessing strategies was synthesized for developers working with SSL in ultrasound.

## 1. Introduction

Automated interpretation of medical ultrasound images is increasingly implemented using deep learning [1]. Deep neural networks (DNNs) achieve strong performance for applications in ultrasound imaging, such as distinguishing benign from malignant liver lesions [2], estimating left ventricular end-diastolic and end-systolic volumes [3], and screening for pneumothorax [4]. Several studies have found that artificial intelligence-based interpretation methods have exhibited strong accuracy across multiple tasks and improved the accessibility of point-of-care ultrasound; however, they struggle to perform well in some disease conditions or when images are poorly acquired [5].

Despite early successes, investigators are limited by the lack of publicly available datasets [6,7]. When available, researchers use private collections of ultrasound examinations, as they may contain far more samples. Given the expense of manual annotation, many are turning to self-supervised learning (SSL) methods to pretrain DNNs using large, unlabelled collections of ultrasound data [8]. These SSL-pretrained backbone DNNs may be fine-tuned for supervised learning tasks of interest.

An important category of SSL methods for computer vision is the joint embedding architecture, which is characterized by training DNNs to produce similar vector representations for pairs of related images. The most common method for retrieving related pairs of images from unlabelled datasets is to apply random transformations (i.e., data augmentation) to an image, producing two distorted views. The choice of random transformations steers the invariance relationships learned by the backbone.

In this study, we proposed and assessed data preprocessing and data augmentation strategies designed to preserve semantic content in medical ultrasound images (Figure 1). We compared handcrafted domain-specific augmentation methods against standard SSL data augmentation practices. We found that ultrasound-specific transformations resulted in the greatest improvement in performance for COVID-19 classification—a diagnostic task—on a public dataset. Experiments also revealed that standard cropping-based augmentation strategies outperformed ultrasound-specific transformations for object classification tasks in lung ultrasound (LU). Lastly, ultrasound-specific semantics-preserving preprocessing was found to be instrumental to the success of pretrained backbones. In summary, our contributions are as follows:Semantics-preserving image preprocessing for SSL in ultrasound;Semantics-preserving data augmentation methods designed for ultrasound images;Comparison of multiple data augmentation strategies for SSL for multiple types of LU tasks;Recommendations for developers working with unlabelled ultrasound datasets.

To our knowledge, this study is the first to quantify the impact of data augmentation methods for SSL with ultrasound. We are hopeful that the results and lessons from this study may contribute to the development of foundation models for medical ultrasound.

## 2. Background

### 2.1. Data Augmentation in Self-Supervised Learning

The joint embedding class of SSL methods is characterized by the minimization of an objective function that, broadly speaking, encourages similarity of related pairs of inputs. Semantically related pairs of images (i.e., positive pairs) are sampled from unlabelled datasets according to a pairwise relationship. If the SSL pairwise relationship is satisfied for samples exhibiting the same class, SSL methods will likely improve the performance of a classifier [9]. Most joint embedding methods rely on data augmentation to define the pairwise relationship. Some studies have used metadata or known relationships between samples to identify related pairs [10,11,12]; however, the availability of such information is rare. The choice of data augmentation transformations is therefore crucial, as it dictates the invariances learned [13]. However, the set of useful invariances differs by the image modality and downstream problem(s) of interest. Despite this, studies continue to espouse a data augmentation pipeline popularized by leading SSL methods, such as SimCLR [14], BYOL [15], Barlow Twins [16], and VICReg [17]. These methods utilized the same pipeline, but with minor hyperparameter variations. The pipeline includes the following transformations: random crops, horizontal reflection, colour jitter, Gaussian blur, and solarization. Hereafter, we refer to this baseline pipeline as StandardAug. Random rotation is an example of a transformation not found in the StandardAug pipeline that represents an important invariance relationship for many tasks in medical imaging. For example, random rotation has been applied in SSL pretraining with magnetic resonance exams of the prostate [18]. Moreover, the authors did not use StandardAug’s Gaussian blur transformation because it may have rendered the images uninterpretable.

### 2.2. Joint Embedding Self-Supervision in Ultrasound

Recent studies have examined the use of joint embedding SSL methods for ultrasound interpretation tasks, such as echocardiogram view classification [19], left ventricle segmentation [20], and breast tumour classification [21]. Some have proposed positive pair sampling schemes customized for ultrasound. The Ultrasound Contrastive Learning (USCL) method and its successors explored contrastive learning methods where the positive pairs were weighted sums of images from the same ultrasound video [22,23,24]. Other methods have studied the use of images from the same video as positive pairs [12,25]. In these studies, the set of transformations was a subset of the StandardAug data augmentation pipeline, occasionally with different hyperparameters. Few studies have proposed ultrasound-specific data augmentation methods for SSL. A recent study by Chen et al. [26] applied BYOL and SimCLR to pretrain 3D convolutional DNNs with specialized data augmentation for lung consolidation detection in ultrasound videos, observing that temporal transformations were contributory to their problem. This study builds on the previous literature by proposing and comparing domain-specific data augmentation and preprocessing methods for multiple types of downstream tasks.

## 3. Materials and Methods

### 3.1. Datasets and Tasks

We assessed the methods in this publication using a combination of public and private data. COVIDx-US is a public COVID-19 LU dataset consisting of 242 publicly sourced videos, acquired from a variety of manufacturers and sites [27]. Each example is annotated with one of the following classes: normal, COVID-19 pneumonia, non-COVID-19 pneumonia, and other lung pathology. Referred to as COVID hereafter, the task is a four-class image classification problem. Since there is no standard test partition, we split the data by patient identifier into training (70%), validation (15%), and test (15%) splits.

The second data source is a private collection of lung ultrasound examinations, and we refer to it as *LUSData*. Access to these data was granted by the research ethics boards at Western University (REB 116838) and the University of Waterloo (REB 43986). LUSData contains videos of parenchymal and pleural views of the lung. A subset of the parenchymal views has labels for the presence of A-lines or B-lines (i.e., the AB classification task). A-lines are reverberation artifacts that indicate normal lung tissue, while B-lines are axial artifacts that indicate fluid or thickness in the lung. A subset of the pleural views is labelled for the presence or absence of pleural effusion (i.e., the PE classification task), which is an accumulation of fluid around the lungs. A small fraction of the parenchymal views in LUSData possess bounding box labels for the pleural line (i.e., the PL object detection task). Most exams in LUSData originated from a local healthcare centre, but a subset were acquired at another centre, which we adopt as an external test set. The labelled examples in the local dataset were split into training (70%), validation (15%), and test (15%) splits by patient. Table 1 provides the video and class counts of LUSData. Further dataset details are in Appendix A. All models in this study are trained on images, instead of on videos. Classification labels apply to every image in the video. However, individual images within each video labelled for the PL task have bounding box annotations.

### 3.2. Semantics-Preserving Preprocessing

The field of view (FOV) in ultrasound images is typically surrounded by burnt-in scan parameters, logos, and other details. We estimated the shape of the FOV and masked out all extraneous graphical entities using ultrasound cleaning software (UltraMask, Deep Breathe Inc., London, ON, Canada, https://www.deepbreathe.ai/products, accessed on 16 June 2025). Semantic information only exists within the FOV of the ultrasound, which typically occupies a fraction of the images. Scaling transformations, such as random cropping, could produce views that largely contain background. Accordingly, we cropped the cleaned images to the smallest rectangle that encapsulates the FOV mask to maximize semantic content in ultrasound images. Figure 2 depicts this semantics-preserving preprocessing workflow. The process was applied to all images in LUSData and COVIDx-US.

### 3.3. Ultrasound-Specific Data Augmentation

Joint embedding SSL is effective when positive pairs contain similar information with respect to downstream tasks [9]. Several SSL studies applied to photographic or medical imaging datasets adopted variations in the StandardAug data augmentation pipeline. The core aim of our study was to determine if semantics-preserving data augmentation would better equip pretrained feature extractors for downstream LU tasks than the commonly applied StandardAug pipeline.

We refer to a data augmentation pipeline as an ordered sequence of transformations, each applied with some probability. For clarity, we assign each transformation an alphanumeric identifier and express a data augmentation pipeline as an ordered sequence of identifiers. The StandardAug pipeline transformations are detailed in Table 2. The table also includes an estimate of the time to transform a single image. Details on how the runtime estimates were calculated are in Appendix B.

We designed the AugUS-O pipeline, which was intended to preserve semantic information in the entire ultrasound FOV while imposing nontrivial differences across invocations. The transformations in AugUS-O are listed below.

B00:**Probe Type Change:** Inspired by Zeng et al.’s work [28], this transformation resamples an ultrasound image according to a different field of view (FOV) shape. Linear FOV shapes are converted to curvilinear shapes, while curvilinear and phased arrays are converted to linear ones.B01:**Convexity Change:** The shape of convex FOVs can vary, depending on the manufacturer, depth, and field of view of the probe. This transformation modifies the FOV shape such that the distance between $x_{1}$ and $x_{2}$ is altered, mimicking a change in $\theta_{0}$.B02:**Wavelet Denoising:** As an alternative to the commonly used Gaussian blur transformation, this transformation denoises an image by thresholding it in wavelet space, according to Birgé and Massart’s method [29].B03:**Contrast-Limited Adaptive Histogram Equalization:** This transformation enhances contrast by applying locally informed equalization [30].B04:**Gamma Correction:** In contrast to standard brightness change transforms, gamma correction applies a nonlinear change in pixel intensity.B05:**Brightness and Contrast Change:** The brightness and contrast of the image are modified by applying a linear transform to the pixel values.B06:**Depth Change Simulation:** Changing the depth controls on an ultrasound probe impacts how far the range of visibility is from the probe. This transformation simulates a change in depth by applying a random zoom while preserving the FOV shape.B07:**Speckle Noise Simulation:** Speckle noise, Gaussian noise, and salt and pepper (S&P) noise are prevalent in ultrasound [31]. This transformation applies Singh et al.’s [32] synthetic speckle noise algorithm to the image.B08:**Gaussian Noise Simulation:** Multiplicative Gaussian noise is independently applied to each pixel.B09:**Salt and Pepper Noise Simulation:** A small, random assortment of pixels is set to black or white.B10:**Horizontal Reflection:** The image is reflected about the central vertical axis.B11:**Rotation and Shift:** The image is rotated and translated by a random angle and vector, respectively.

Refer to Figure 3 for a visual example of each transformation in AugUS-O. Algorithmic details and parameter settings for the StandardAug and AugUS-O pipelines are in Appendix C and Appendix D, respectively. As is common in stochastic data augmentation, each transformation was applied with some probability. Table 3 gives the entire sequence of transformations and the probability with which each is applied. Visuals of positive pairs produced using the StandardAug and AugUS-O augmentation pipelines can be found in Figure 4a and Figure 4b, respectively.

We conducted an informal assessment of the similarity of positive pairs. Positive pairs were produced for 50 randomly sampled images, using both the StandardAug and the AugUS-O pipelines. The pairs were presented in random order to one of the authors, who is an expert in point-of-care ultrasound. They were aware of the two pipelines but were not told which pipeline produced each pair. The expert was asked to mark the pairs they believed conveyed the same clinical impression. We observed that 58% of pairs produced with the StandardAug pipeline were marked as similar, whereas 70% of the AugUS-O pairs were marked as similar. While not conclusive, this manual evaluation added credence to the semantics-preserving intention of the design.

### 3.4. Discovering Semantically Contributory Transformations

A major aim of this work was to explore the utility of various data augmentation schemes during pretraining. As such, we conducted leave-one-out analysis for each of the StandardAug and AugUS pipelines to estimate the impact of each transformation on the models’ ability to solve downstream classification tasks. We pretrained separate models on the unlabelled portion of LUSData, using an altered version of a pipeline with one transformation omitted. We then conducted 10-fold cross-validation on the LUSData training set for downstream classification tasks for each pretrained model. The median cross-validation test performance for each model pretrained using an ablated pipeline was compared to a baseline model that was pretrained with the entire pipeline. The experiment was conducted for both the StandardAug and AugUS pipelines. Any transformations that, when omitted, resulted in worsened performance on either AB or PE were deemed contributory.

### 3.5. Training Protocols

We adopted the MobileNetV3Small architecture [33] for all experiments in this study and pretrained using the SimCLR method [14]. MobileNetV3Small was chosen due to its real-time inference capability on mobile devices and its use in prior work by VanBerlo et al.  for similar tasks [34]. Local inference on edge devices is especially important in point-of-care ultrasound imaging, as modern ultrasound devices are used in austere settings with limited internet access. The SimCLR projector was a 2-layer multilayer perceptron with 576 nodes per layer. Images were resized to 128×128 pixels prior to the forward pass, which is consistent with prior work for similar tasks [34]. Unless otherwise stated, backbones (i.e., feature extractors) were initialized using ImageNet-pretrained weights [35] and were pretrained using the LARS optimizer [36] with a batch size of 1024, a base learning rate of 0.2, and a linear warmup with a cosine decay schedule. Pretraining was conducted for 3 epochs with 0.3 warmup epochs for LUSData, and 100 epochs with 10 warmup epochs for COVIDx-US.

To conduct supervised evaluation, a perceptron classification head was appended to the final pooling layer of the backbone. Classifiers were trained using stochastic gradient descent with a momentum of 0.9 and a batch size of 512. The learning rates for the backbone and head were 0.002 and 0.02, respectively; each was annealed according to a cosine decay schedule. Training was conducted for 10 epochs on LUSData and 30 epochs on COVIDx-US. Unless otherwise stated, the weights corresponding to the epoch with the lowest validation loss were retained for test set evaluation.

Although this study focused on classification tasks, we also evaluated backbones on the PL object detection task using the Single Shot Detector (SSD) method [37]. SSL-pretrained backbones were used as the convolutional feature extractor. Architectural and training details for SSD are in Appendix E.

Self-supervised pretraining was conducted using virtual machines equipped with an Intel E5-2683 v4 Broadwell CPU at 2.1 GHz and 2 Nvidia Tesla P100 GPUs each with 12 GB of VRAM. Supervised training was conducted using the same hardware, except with a single GPU. Source code for the experiments and transformations is available in a public GitHub repository (https://github.com/bvanberl/aug_us_ssl, release v1).

## 4. Results

### 4.1. Transformation Leave-One-Out Analysis

Leave-one-out analysis was conducted to discover which transformations in each of the StandardAug and AugUS-O pipelines were contributory to downstream task performance. We pretrained backbones using versions of each pipeline with one transformation omitted. The private LUSData training set was split by patient into 10 disjoint subsets. For each pretrained backbone, 10-fold cross-validation was conducted to obtain estimates of the performance of linear classifiers trained on its output feature vectors. The maximum validation area under the receiver operating characteristic curve (AUC) across epochs was recorded. Omitted transformations that resulted in statistically significant lower validation AUC for either the AB or PE task were included in a third pipeline.

We conducted statistical testing to compare each of the StandardAug and AugUS-O pipelines, and for each of the AB and PE tasks (described in Section 3.1). Friedman’s test for multiple comparisons [38] was conducted, with significance level 0.05. When significant differences were found, we performed the Wilcoxon Signed-Rank Test [39] to compare the test AUCs from each ablated model to the baseline’s test AUC valuess. To control for false positives, the Holm-Bonferroni correction [40] was applied to keep the family-wise significance level at 0.05.

Table 4 details the results of the leave-one-out analysis. Friedman’s test detected differences in performance on both the AB and PE tasks when pretrained using the StandardAug pipeline, but only the AB task exhibited differences when pretrained with the AugUS-O pipeline. As shown in Table 4, the set of transformations that exhibited statistically significant reductions in test AUC for at least one task when excluded were crop and resize (A00), colour jitter (A02), CLAHE (B03), and rotation and shift (B11). Appendix F provides all test statistics from this investigation. Of note is the sharp decrease in performance without the random crop and resize (A00), indicating that it is a critical transformation.

Using these transformations, we constructed a distilled pipeline that consists only of the above transformations. Referred hereafter to as AugUS-D, the pipeline is expressed as the following sequence: [B03, A02, B11, A00]. Figure 4c provides some examples of positive pairs produced with AugUS-D. For more examples of pairs produced by each pipeline, see Appendix G.

### 4.2. Object Classification Task Evaluation

The StandardAug, AugUS-O, and AugUS-D pipelines were compared in terms of their performance on multiple downstream tasks. Model backbones were pretrained using each of the data augmentation pipelines on the union of the unlabelled and training sets in LUSData. Linear evaluation and fine-tuning experiments were performed according to the procedure explained in Section 3.5. In this section, we present results on the two object classification tasks: A-line vs. B-line classification (AB) and pleural effusion classification (PE).

Linear classifiers indicate the usefulness of pretrained backbones, as the only trainable weights for supervised learning are those belonging to the perceptron head. Table 5 reports the test set performance of linear classifiers for each task and data augmentation pipeline. On the private dataset, the AugUS-D and StandardAug pipelines performed comparably well on the AB task. AugUS-D attained greater performance metrics than the StandardAug pipeline on PE. To provide a visual perspective on linear classifier performance, we produced two-dimensional t-distributed Stochastic Neighbour Embeddings (t-SNEs) of the feature vectors outputted by pretrained backbones [41]. Shown in Figure 5, the separability of the visual representations is consistent with linear classifier performance.

We fine-tuned the pretrained models, allowing the backbone’s weights to be trainable in addition to the model head. Table 5 gives the test set performance of the fine-tuned classifiers. We observed similar performance differences among the different augmentation pipelines, but note some additional findings. The model pretrained using AugUS-O on LUSData performed comparably against the other pipelines on AB but exhibited extremely poor performance on the PE test set. Although it may appear that this model may have overfit to the training set, examination of training metrics revealed that training and validation metrics were close, with validation set AUC having been evaluated as 0.861. Nonetheless, fine-tuned models that were pretrained with the StandardAug and AugUS-O pipelines yielded strong performance on both tasks in LUSData.

Linear and fine-tuned classifiers for the AB and PE tasks were also evaluated on the external portion of the LUSData test set. Metrics for external test set predictions are provided in Table 6. Most classifiers exhibited degraded performance on external data, compared to the local test set. Overall, the relative performance of the classifiers on the external test set was reflective of their performance on local test data. SimCLR-pretrained linear and fine-tuned classifiers achieved greater performance than those initialized with ImageNet-pretrained or randomly sampled weights on external AB and PE test sets. Note the difference in the manufacturer and probe type distributions between the local and external datasets (see Table A1)—there was a far greater proportion of videos gathered by linear probes and Philips-manufactured devices in the external test set. Unlike the local test evaluation, the network trained from scratch performed comparably on AB to the SimCLR-pretrained models that utilized the AugUS-O and AugUS-D pipelines. However, the SimCLR-pretrained model that utilized the StandardAug pipeline achieved the greatest AB test AUC by a margin of 0.029 among the fine-tuned models. On the PE task, the classifier originating from the same pretrained model that utilized StandardAug achieved the greatest AUC by a margin of 0.019 among the fine-tuned models. In the linear classifier setting, models pretrained using StandardAug and AugUS-D performed comparably well on the external test set, achieving greater AUC than the model pretrained using AugUS-O. Similar to the local test set, the pretrained models that incorporated the StandardAug and AugUS-D pipeline achieved the greatest test AUC, while the pretrained model that utilized AugUS-O performed the worst.

Although MobileNetV3Small was the backbone architecture used for these experiments, we repeated the above evaluations using the more commonly employed ResNet18 architecture [42]. Similar trends were observed regarding the greater test performance attained by models pretrained with cropping-based pipelines. However, the fine-tuned models greatly overfit, likely due to ResNet18’s much greater capacity. The ResNet18 models that achieved the greatest test performance were the linear classifiers trained on frozen backbones. Notably, the trend persisted when evaluating on external test data. Detailed results for ResNet18 can be found in Appendix H.

### 4.3. Diagnostic Classification Task Evaluation

Models pretrained on LUSData were also evaluated on the COVID-19 multi-class problem (COVID). Unlike the AB and PE tasks, COVID is a diagnostic task that involves global image understanding, as the relationship between objects is pertinent. Multiple findings on lung ultrasound have been observed in the context of COVID-19 pneumonia, including B-lines, pleural line abnormalities, and consolidation [43].

Linear classifiers were trained on the COVID training set and evaluated on the COVID test set. As shown in Table 7, AugUS-O was observed to have the greatest test multiclass AUC, which was considered the primary metric of interest. Looking at the t-SNE visualizations in Figure 5, AugUS-O corresponds to the only visualization where the representations for the COVID-19 Pneumonia and non-COVID-19 Pneumonia classes are clustered together.

Table 7 provides test metrics for fine-tuned COVID classifiers. Again, AugUS-O exhibited the best performance. Moreover, fine-tuned models generally performed worse than the linear classifiers trained on feature vectors from SSL-pretrained models; they likely suffered from overfitting, as COVIDx-US is a smaller dataset.

Unlike the AB and PE tasks, models trained for COVID were pretrained using the LUSData dataset. We repeated the linear classification and fine-tuning experiments using models pretrained on the COVIDx-ultrasound training set. Table 7 reports the results of linear and fine-tuning evaluations. As was observed for backbones pretrained on LUSData, pretraining with the AugUS-O pipeline resulted in the greatest test set AUC.

The trends observed for the COVID task are different than those observed for the AB and PE tasks. Regardless of the augmentation pipeline, SimCLR-pretrained weights resulted in better performance than ImageNet-pretrained or random weight initialization. On object classification tasks, models pretrained using the StandardAug and AugUS-D pipelines performed the best. However, on the diagnostic COVID task, AugUS-O performed best. Recall that AugUS-O was designed to retain semantic information, while both StandardAug and AugUS-D contain the very impactful crop and resize (C&R) transform that can obscure large portions of the image. Object classification tasks require scale invariance, which is enforced by applying C&R during SSL pretraining. Diagnostic tasks, on the other hand, require global image context for interpreters to make a decision, which is preserved best by the AugUS-O pipeline.

### 4.4. Object Detection Task Evaluation

Recall that the PL task is an object detection problem geared toward localizing the pleural line. We evaluated the pretrained models on PL to explore whether the trends observed for object-centric LU classification tasks would hold for an object detection task, where locality understanding is explicit. We considered two evaluation settings: one in which the pretrained backbone’s weights were held constant, and another in which the backbone’s weights were trainable. Table 8 reports the average precision at a 50% intersection over union threshold (AP@50) evaluated on the LUSData test set. When the backbone weights were frozen, SimCLR pretraining with AugUS-O resulted in the greatest test AP@50. These trends differed from the results observed for AB and PE classification, which both require object recognition. We speculate that aggressive cropping during the pretraining phase likely produced positive pairs where one image contained a pleural line while the other did not, which we believe would make it difficult to learn representations for pleural line objects when pretraining with the StandardAug or AugUS-D pipelines. The performance of the frozen pretrained backbones was strong overall, considering the low capacity of the backbone and the small, narrow shape of pleural line objects. When fine-tuning end-to-end, SimCLR pretraining with AugUS-D resulted in the greatest test AP@50.

### 4.5. Label Efficiency Assessment

Experiments were conducted to test the robustness of pretrained models in settings where few labelled samples are available. The experiment was conducted only for the AB and PE classification tasks because there were enough unique videos and patients in the training set to create several disjoint subsets. Backbones were fine-tuned on 20 subsets of approximately 5% of the LUSData training set, split by patient, yielding 20 performance estimates for low-label settings. Splitting was conducted at the patient level to heighten the difficulty of the task and to limit dependence between subsets. Baseline estimates without SSL pretraining were obtained via initialization with random weights and with ImageNet-pretrained weights, resulting in five different performance conditions. Figure 6 displays boxplots for test AUC distributions under each condition. Friedman’s test indicated that there were significant differences among the median test AUC across conditions, for both the AB and PE tasks. Post-hoc Wilcoxon Signed-Rank Tests were then conducted for each pair of conditions, using the Bonferroni correction with a family-wise error rate of α=0.05. The median test AUCs of SimCLR-pretrained models were significantly greater than those initialized with random or ImageNet-pretrained weights for both the AB and PE tasks. All medians were significantly different for AB, except for the SimCLR-pretrained models using the StandardAug and AugUS-D pipelines, which achieved the greatest performance. Notably, these pipelines both utilize the crop and resize transformation. No significant differences were observed between any of the SimCLR-pretrained models for PE. Appendix I provides the test statistics for the above comparisons.

### 4.6. Impact of Semantics-Preserving Preprocessing

As outlined in Section 3.2, all ultrasound images were cropped to the smallest rectangle enclosing the FOV because the areas outside the FOV are bereft of information. Since pipelines containing the crop and resize transformation (C&R) would be more likely to result in positive pairs that do not cover the FOV, it was hypothesized that cropping to the FOV as a preprocessing step would result in stronger pretrained backbones. To investigate the effect of this semantics-preserving preprocessing step, we pretrained backbones on LUSData using each data augmentation pipeline and evaluated them on the AB, AB, and COVID tasks. Table 9 compares the performance of each backbone with and without the preprocessing step. Performance on the AB task did not change. However, test AUC on both the PE and COVID tasks was consistently lower when the semantics-preserving preprocessing was not applied. Note that greatly less labelled data is available for PE and COVID than for AB. Based on these experiments, FOV cropping is a valuable semantics-preserving preprocessing step for multiple LU classification tasks.

### 4.7. Impact of the Cropping in Object Classification Tasks

The leave-one-out analysis for transformations exhibited the striking finding that crop and resize (C&R) was the most effective transformation in the StandardAug pipeline for the two object classification tasks: AB and PE. Moreover, both pipelines containing C&R resulted in the greatest downstream test performance on AB and PE. Ordinarily, crops are taken at random locations in an image, with areas between 8% and 100% of the original image’s area. Aggressive crops can create situations in which positive pairs do not contain the same objects of interest. Figure 7 shows how C&R could produce positive pairs with different semantic content. Despite this, the results indicated that pipelines containing C&R led to improved performance for the object-centric AB and PE tasks. The exceptional influence of C&R warranted further investigation into its hyperparameters.

We investigated the impact of the minimum crop area, *c*, as a hyperparameter. Models were pretrained with the AugUS-D pipeline, using values for *c* in the range [0.05,0.9]. Linear evaluation was conducted for the AB and PE tasks. As shown in Figure 8, smaller values of *c* yielded better performance, peaking at c≈0.1. The default assignment of c=0.08 was already a reasonable choice for these two tasks. Additional experiments elucidating the effects of C&R hyperparameters can be found in Appendix J.

Another concern with C&R is that it could result in crops covering the black background on images with a convex FOV. Despite the semantics-preserving preprocessing (described in Figure 2), the top left and right corners of such images provide no information. To characterize the robustness of pretraining under these circumstances, we repeated the experiments sweeping over c∈[0.05,0.9] but first applied the probe type change transformation (i.e., B00) to every convex FOV. Thus, all inputs to the model were linear FOVs devoid of non-semantic background. A by-product of this transformation is that the near fields of convex images are horizontally stretched. As seen in Figure 8, this change resulted in a slight decrease in performance for both the AB and PE tasks. Evidently, the detriment of spatial distortion outweighed the benefit of guaranteeing that crops were positioned over semantic regions.

Overall, it is clear that aggressive C&R is beneficial for distinguishing between A-lines and B-lines and detecting pleural effusions on LU. Both are object-centric classification tasks. Even though some crops may not contain the object, the backbone would be exposed to several paired instances of transformed portions of objects during pretraining, potentially facilitating texture and shape recognition. Conversely, solving diagnostic tasks such as COVID requires a holistic assessment of the FOV, wherein the co-occurrence of objects is contributory to the overall impression.

## 5. Conclusions

This study proposed and evaluated data augmentation and preprocessing strategies for self-supervised learning in ultrasound. A commonly employed baseline pipeline (StandardAug) was compared to a handcrafted semantics-preserving pipeline (AugUS-O) and a hybrid pipeline (AugUS-D) composed from the first two. Evaluation of LU interpretation tasks revealed a dichotomy between the utility of the pipelines. Pipelines featuring the cropping transformation (StandardAug and AugUS-D) were most useful for object classification and detection tasks in LU. On the other hand, AugUS-O—designed to preserve semantics in LU—resulted in the greatest performance on a diagnostic task that required global context. Additionally, ultrasound field of view cropping was found to be a beneficial preprocessing step for multiple LU classification tasks, regardless of the data augmentation strategy.

Based on the results of this study, we provide guidance for machine learning practitioners seeking to apply self-supervised pretraining for tasks in ultrasound imaging. First, developers should use semantics-preserving preprocessing during pretraining that crops images to the bounds of the ultrasound FOV. When considering data augmentation strategies for pretraining, semantics-preserving transformations should be considered for tasks requiring holistic interpretation of images, while cropping-based transformations should be leveraged for object-centric downstream tasks.

Some limitations are acknowledged in this study. For example, SimCLR was the only SSL objective that was investigated, and all downstream tasks were confined to the lung. Moreover, some of the transformations introduced in this work constitute computationally expensive preprocessing steps, as they are applied with nonzero probability to each image. Lastly, while AugUS-O was composed of several transformations, we acknowledge that it does not encapsulate all possible transformations that could preserve semantic information in ultrasound images.

Future work should apply this study’s methods to assess the impact of data augmentation pipelines for ultrasound diagnostic tasks outside of the lung and for other SSL methods. Future studies could also compare data augmentation strategies for localization and segmentation downstream tasks in ultrasound.

## Figures and Tables

**Figure 1 bioengineering-12-00855-f001:**
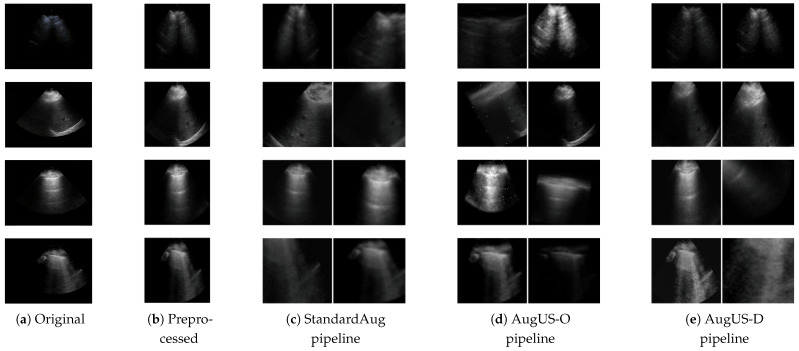
Examples of the preprocessing and data augmentation methods in this study. (**a**) Original images are from ultrasound exams. (**b**) Semantics-preserving preprocessing is applied to crop out areas external to the field of view. (**c**) The StandardAug pipeline is a commonly employed data augmentation pipeline in self-supervised learning. (**d**) The AugUS-O pipeline was designed to preserve semantic content in ultrasound images. (**e**) AugUS-D is a hybrid pipeline whose construction was informed by empirical investigations into the StandardAug and AugUS-O pipelines.

**Figure 2 bioengineering-12-00855-f002:**
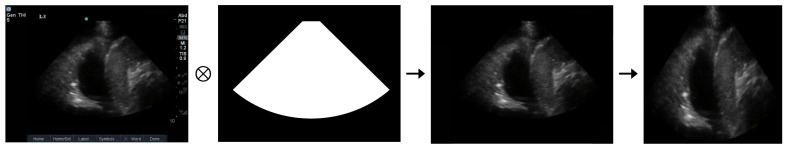
Raw ultrasound images are preprocessed by performing an element-wise multiplication (⊗) of the raw image with a binary mask that preserves only the field of view, then cropped according to the bounds of the field of view.

**Figure 3 bioengineering-12-00855-f003:**
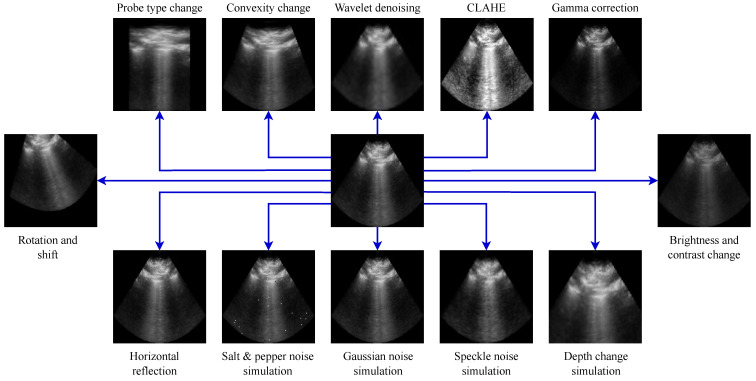
Examples of ultrasound-specific data augmentation transformations applied to the same ultrasound image.

**Figure 4 bioengineering-12-00855-f004:**
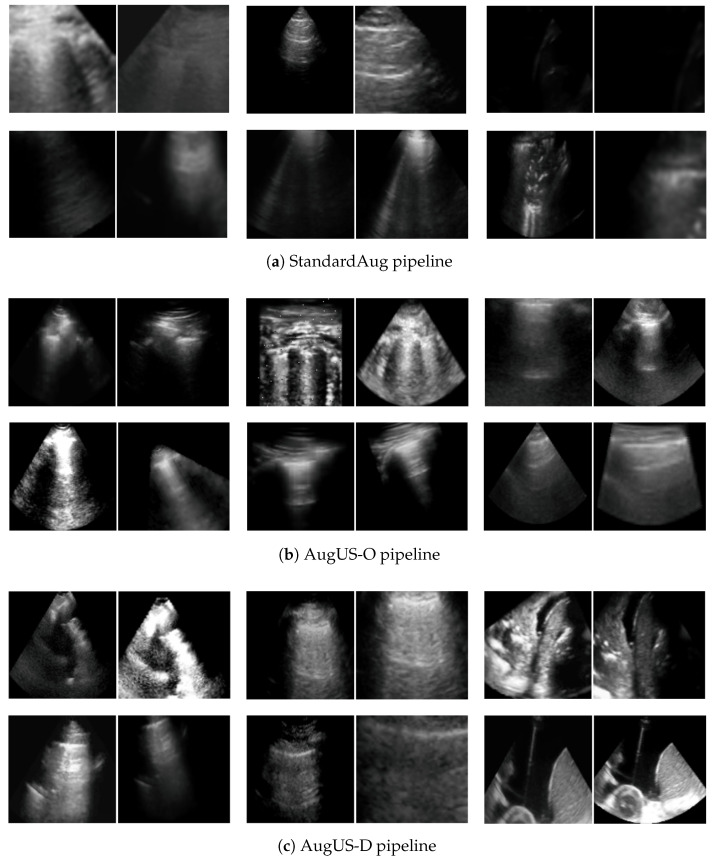
Examples of positive pairs produced using each of the (**a**) StandardAug, (**b**) AugUS-O, and (**c**) AugUS-D data augmentation pipelines.

**Figure 5 bioengineering-12-00855-f005:**
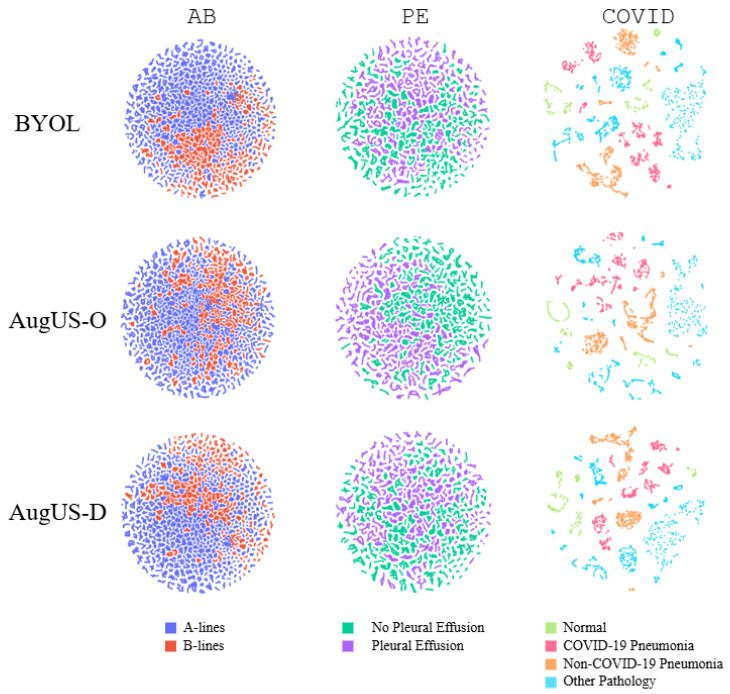
Two-dimensional t-distributed Stochastic Neighbour Embeddings (t-SNEs) for test set feature vectors produced by SimCLR-pretrained backbones, for all tasks and data augmentation pipelines.

**Figure 6 bioengineering-12-00855-f006:**
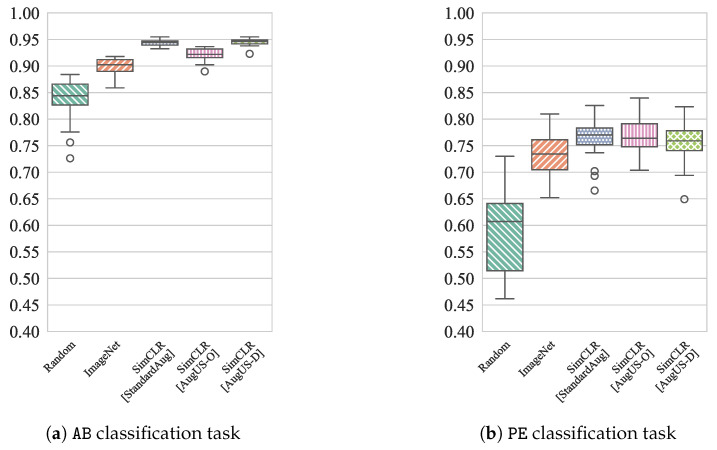
Distribution of test AUC for classifiers trained on disjoint subsets of 5% of the patients in the training partition of LUSData for (**a**) the AB task and (**b**) the PE task.

**Figure 7 bioengineering-12-00855-f007:**
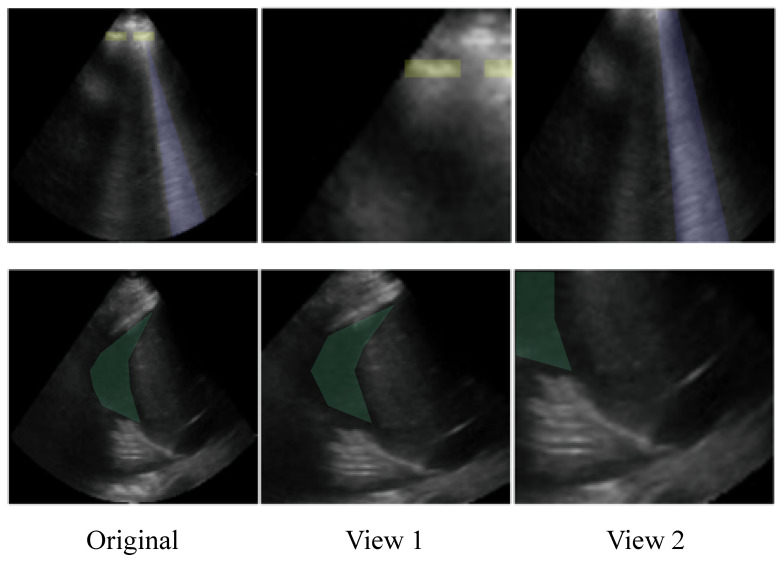
Examples of how the random crop and resize transformation (A00) can reduce semantic information. Original images are on the left, and two random crops of the image are on the right. **Top:** The original image contains a B-line (purple), which is visible in View 2 but not in View 1. The original image also contains instances of the pleural line (yellow) which are visible in View 1 but not in View 2. **Bottom:** The original image contains a pleural effusion (green), which is visible in View 1 but largely obscured in View 2.

**Figure 8 bioengineering-12-00855-f008:**
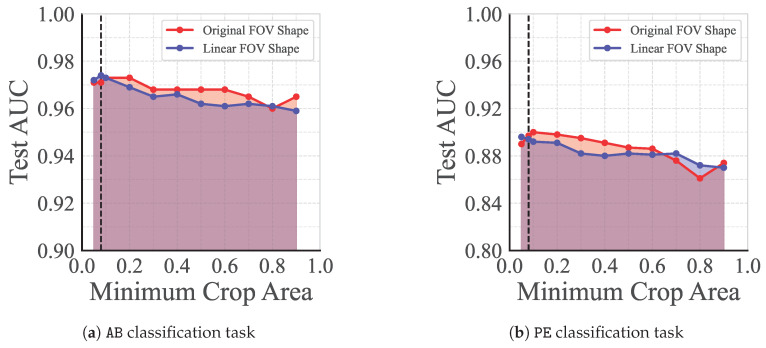
Test set AUC for linear classifiers trained on the representations outputted by pretrained backbones, for (**a**) the AB task and (**b**) the PE task. Each backbone was pretrained using AugUS-D with different values for the minimum crop area, *c*. Results are provided for models pretrained with the original ultrasound FOV, along with images transformed to linear field of view (FOV) only. The dashed line indicates the default value of c=0.08.

**Table 1 bioengineering-12-00855-t001:** Breakdown of the unlabelled, training, validation, and test sets in the private dataset. For each split, we indicate the number of distinct patients, videos, and images. x/y indicates the number of labelled videos in the negative and positive class for each binary classification task. For the PL task, we indicate the number of videos with frame-level bounding box annotations.

	Local				External
	Unlabelled	Train	Validation	Test	Test
Patients	5571	1702	364	364	168
Videos	59,309	5679	1184	1249	925
Images	1.3 × 10^7^	1.2 × 10^6^	2.5 × 10^5^	2.6 × 10^5^	1.1 × 10^5^
AB labels	N/A	2067/999	459/178	458/221	286/327
PE labels	N/A	789/762	176/142	162/158	68/110
PL labels	N/A	200	39	45	0

**Table 2 bioengineering-12-00855-t002:** The sequence of transformations in the StandardAug data augmentation pipeline.

Identifier	Probability	Transformation	Time [ms]
A00	1.0	Crop and resize	0.29
A01	0.5	Horizontal reflection	0.08
A02	0.8	Colour jitter.	2.40
A03	0.2	Conversion to grayscale	0.19
A04	0.5	Gaussian blur	0.74
A05	0.1	Solarization	0.15

**Table 3 bioengineering-12-00855-t003:** The sequence of transformations in the ultrasound-specific augmentation pipeline.

Identifier	Probability	Transformation	Time [ms]
B00	0.3	Probe type change	2.25
B01	0.75	Convexity change	1.92
B02	0.5	Wavelet denoising	5.00
B03	0.2	CLAHE ^†^	4.64
B04	0.5	Gamma correction	0.52
B05	0.5	Brightness and contrast change	0.49
B06	0.5	Depth change simulation	1.76
B07	0.333	Speckle noise simulation	3.69
B08	0.333	Gaussian noise	0.28
B09	0.1	Salt and pepper noise	0.18
B10	0.5	Horizontal reflection	0.19
B11	0.5	Rotation and shift	1.42

^†^ Contrast-limited adaptive histogram equalization.

**Table 4 bioengineering-12-00855-t004:** A comparison of ablated versions of the StandardAug and AugUS-O pipeline, with one excluded transformation versus the original pipelines. Models were pretrained on the LUSData unlabelled set and evaluated on two downstream classification tasks—AB and PE. Performance is expressed as mean and median test area under the receiver operating characteristic curve (AUC) from 10-fold cross-validation achieved by a linear classifier trained on the feature vectors of a frozen backbone.

Pipeline	Omitted	AB		PE	
		Mean (std)	Median	Mean	Median
StandardAug	None	0.978(0.007)	0.978	0.852(0.040)	0.845
	A00	0.864(0.022)	0.873 ^†^	0.695(0.050)	0.707 ^†^
	A01	0.976(0.006)	0.974	0.848(0.046)	0.856
	A02	0.975(0.007)	0.975 ^†^	0.840(0.046)	0.842 ^†^
	A03	0.978(0.007)	0.978	0.849(0.044)	0.846
	A04	0.976(0.007)	0.975	0.840(0.041)	0.842
	A05	0.977(0.007)	0.977	0.851(0.041)	0.853
AugUS-O	None	0.956(0.013)	0.959	0.828(0.030)	0.837
	B00	0.958(0.011)	0.957	0.831(0.034)	0.839
	B01	0.952(0.016)	0.952	0.835(0.027)	0.838
	B02	0.965(0.011)	0.967 ^§^	0.840(0.032)	0.851
	B03	0.950(0.011)	0.951 ^†^	0.825(0.028)	0.827
	B04	0.957(0.013)	0.958	0.831(0.034)	0.836
	B05	0.953(0.014)	0.952	0.839(0.024)	0.845
	B06	0.961(0.009)	0.959	0.829(0.037)	0.833
	B07	0.959(0.012)	0.960	0.838(0.035)	0.856
	B08	0.961(0.013)	0.966 ^§^	0.834(0.027)	0.849
	B09	0.962(0.012)	0.967 ^§^	0.838(0.030)	0.845
	B10	0.956(0.011)	0.959	0.826(0.035)	0.838
	B11	0.937(0.020)	0.939 ^†^	0.825(0.028)	0.823

^†^ Median is significantly less than baseline, where no transformations were omitted. ^§^ Median is significantly greater than baseline, where no transformations were omitted.

**Table 5 bioengineering-12-00855-t005:** Test set performance for linear classification (LC) and fine-tuning (FT) experiments with the AB and PE tasks. Binary metrics are averages across classes. The best observed metrics in each experimental setting are in **boldface**.

Train Setting	Task	Initial Weights	Pipeline	Accuracy	Precision	Recall	AUC
LC	AB	SimCLR	StandardAug	0.932	0.951	0.819	0.970
		SimCLR	AugUS-O	0.910	0.939	0.756	0.953
		SimCLR	AugUS-D	0.931	0.947	0.820	0.971
		ImageNet	-	0.898	0.894	0.758	0.949
	PE	SimCLR	StandardAug	0.782	0.769	0.787	0.881
		SimCLR	AugUS-O	0.795	0.796	0.776	0.853
		SimCLR	AugUS-D	0.800	0.798	0.782	0.886
		ImageNet	-	0.779	0.756	0.804	0.864
FT	AB	SimCLR	StandardAug	0.941	0.951	0.850	0.970
		SimCLR	AugUS-O	0.939	0.938	0.859	0.968
		SimCLR	AugUS-D	0.931	0.960	0.809	0.962
		Random	-	0.883	0.794	0.834	0.938
		ImageNet	-	0.911	0.872	0.830	0.953
	PE	SimCLR	StandardAug	0.766	0.713	0.863	0.882
		SimCLR	AugUS-O	0.487	0.479	0.685	0.557
		SimCLR	AugUS-D	0.802	0.782	0.818	0.884
		Random	-	0.703	0.733	0.607	0.767
		ImageNet	-	0.708	0.640	0.907	0.845

**Table 6 bioengineering-12-00855-t006:** External test set metrics for linear classification (LC) and fine-tuning (FT) experiments with the AB and PE tasks. Binary metrics are averages across classes. The best observed metrics in each experimental setting are in **boldface**.

Train Setting	Task	Initial Weights	Pipeline	Accuracy	Precision	Recall	AUC
LC	AB	SimCLR	StandardAug	0.749	0.956	0.555	0.868
		SimCLR	AugUS-O	0.689	0.927	0.453	0.810
		SimCLR	AugUS-D	0.726	0.922	0.531	0.859
		ImageNet	-	0.643	0.872	0.389	0.770
	PE	SimCLR	StandardAug	0.794	0.835	0.843	0.880
		SimCLR	AugUS-O	0.784	0.916	0.728	0.870
		SimCLR	AugUS-D	0.806	0.877	0.809	0.887
		ImageNet	-	0.758	0.836	0.771	0.840
FT	AB	SimCLR	StandardAug	0.751	0.961	0.556	0.883
		SimCLR	AugUS-O	0.734	0.960	0.523	0.850
		SimCLR	AugUS-D	0.712	0.934	0.500	0.854
		Random	-	0.748	0.885	0.606	0.853
		ImageNet	-	0.718	0.903	0.527	0.814
	PE	SimCLR	StandardAug	0.805	0.838	0.861	0.898
		SimCLR	AugUS-O	0.572	0.649	0.717	0.536
		SimCLR	AugUS-D	0.800	0.904	0.768	0.879
		Random	-	0.700	0.850	0.643	0.804
		ImageNet	-	0.776	0.775	0.914	0.840

**Table 7 bioengineering-12-00855-t007:** Test set performance for linear classification (LC) and fine-tuning (FT) experiments with the COVID task. Binary metrics are averages across classes. The best observed metrics in each experimental setting are in **boldface**.

Train Setting	Pretraining Dataset	Initial Weights	Pipeline	Accuracy	Precision	Recall	AUC
LC	LUSData	SimCLR	StandardAug	0.454	0.371	0.413	0.784
		SimCLR	AugUS-O	0.560	0.431	0.513	0.836
		SimCLR	AugUS-D	0.487	0.348	0.447	0.713
	COVIDx-US	SimCLR	StandardAug	0.498	0.582	0.501	0.781
		SimCLR	AugUS-O	0.557	0.506	0.543	0.820
		SimCLR	AugUS-D	0.540	0.400	0.491	0.760
	-	ImageNet	-	0.503	0.304	0.451	0.699
FT	LUSData	SimCLR	StandardAug	0.381	0.259	0.365	0.753
		SimCLR	AugUS-O	0.557	0.428	0.509	0.836
		SimCLR	AugUS-D	0.465	0.321	0.430	0.744
	COVIDx-US	SimCLR	StandardAug	0.450	0.540	0.464	0.770
		SimCLR	AugUS-O	0.517	0.483	0.510	0.814
		SimCLR	AugUS-D	0.526	0.384	0.479	0.672
	-	Random	-	0.423	0.327	0.401	0.534
	-	ImageNet	-	0.502	0.305	0.457	0.698

**Table 8 bioengineering-12-00855-t008:** LUSData local test set AP@50 for the PL task observed for SSD models whose backbones were pretrained using different data augmentation pipelines. **Boldface** values indicate top performance.

Backbone	Initial Weights	Pipeline	AP@50
Frozen	SimCLR	StandardAug	0.228
	SimCLR	AugUS-O	0.255
	SimCLR	AugUS-D	0.194
	Random	-	0.041
	ImageNet	-	0.127
Trainable	SimCLR	StandardAug	0.316
	SimCLR	AugUS-O	0.332
	SimCLR	AugUS-D	0.351
	Random	-	0.308
	ImageNet	-	0.310

**Table 9 bioengineering-12-00855-t009:** Test set AUC for SimCLR-pretrained models with (✓) and without (✗) semantics-preserving preprocessing. Results are reported for linear classifiers and fine-tuned models.

		Linear Classifier		Fine-Tuned	
Task	Pipeline/Preprocessing	✗	✓	✗	✓
AB	StandardAug	0.971	0.970	0.971	0.970
	AugUS-O	0.950	0.953	0.926	0.968
	AugUS-D	0.971	0.971	0.961	0.962
PE	StandardAug	0.873	0.893	0.869	0.882
	AugUS-O	0.846	0.865	0.522	0.557
	AugUS-D	0.867	0.897	0.864	0.884
COVID	StandardAug	0.742	0.784	0.724	0.753
	AugUS-O	0.793	0.836	0.805	0.836
	AugUS-D	0.585	0.713	0.737	0.744

## Data Availability

The LUSData dataset is not readily available due to restrictions imposed by the data owner. It proprietary and thus cannot be shared. The COVIDxUS dataset is available to the public in the COVID-US repository at https://github.com/nrc-cnrc/COVID-US, as presented by Ebadi et al.  [27].

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
