# Peer review of "The Efficacy of Semantics-Preserving Transformations in Self-Supervised Learning for Medical Ultrasound"

_bioengineering, 2025, doi:10.3390/bioengineering12080855_

Round 1

Reviewer 1 Report

Comments and Suggestions for Authors
  • Please add a visual diagram to illustrate the three augmentation pipelines and how they differ.

  • Include quantitative metrics or a performance comparison table to show improvements across the tasks more clearly.

  • Clarify what specific transformations are included in the “semantic-preserving” pipeline.

  • Expand on the methodology used to select and evaluate the “distilled set” of transformations.

  • Provide more detailed guidance or a checklist at the end for developers applying SSL in ultrasound imaging.

Author Response

Comments 1: Please add a visual diagram to illustrate the three augmentation pipelines and how they differ.

Response 1: We respectfully point the reviewer to Figure 1, which illustrates examples of applying each of the three data augmentation pipelines. Moreover, we have carefully described each pipeline and their constituent transformations in Section 3.3, Section 4.1, Appendix C, and Appendix D. 

Comments 2: Include quantitative metrics or a performance comparison table to show improvements across the tasks more clearly.

Response 2: We thank the reviewer for their comment. We refer the reviewer to the following tables, which provide clear classification metrics for each pipeline and task, along with indicators corresponding to the best-performing pipelines:

  • Table 5: provides detailed classification metrics on the local LUSData test set for each pipeline across the A-lines versus B-lines task and the pleural effusion classification task, for the MobileNetV3Small architecture
  • Table 6: provides detailed classification metrics on the external LUSData test set for each pipeline across the A-lines versus B-lines task and the pleural effusion classification task, for the MobileNetV3Small architecture
  • Table 7: provides detailed classification metrics on the local LUSData test set for each pipeline on the COVID-19 classification task, for the MobileNetV3Small architecture
  • Table 8: provides average precision evaluated at 50% IoU on the local LUSData test set for each pipeline on the pleural line object detection task, for the MobileNetV3Small architecture
  • Table 9: Table 5: provides AUC on the local LUSData test set with and without semantics-preserving preprocessing. We provide results for each pipeline on the A-lines versus B-lines test set, the pleural effusion test set, and the COVID-19 classification test set
  • Table A6: provides detailed classification metrics on the local LUSData test set for each pipeline across the A-lines versus B-lines task and the pleural effusion classification task, for the ResNet18 architecture
  • Table A7: provides detailed classification metrics on the external LUSData test set for each pipeline across the A-lines versus B-lines task and the pleural effusion classification task, for the ResNet18 architecture. Note that we added a sentence in Appendix H ensuring that we referenced this table in the main text (lines 736-737).

Comments 3: Clarify what specific transformations are included in the “semantic-preserving” pipeline.

Response 3: This information is provided in Section 3.3 (lines 150-180). Appendix D contains very detailed explanations of the transformations in this pipeline (lines 546-638).

Comments 4: Expand on the methodology used to select and evaluate the “distilled set” of transformations.

Response 4: We conducted a leave-one-out analysis for each of the StandardAug and AugUS-O pipelines to determine which transformations that, when omitted from the pipeline, resulted in worse performance on the validation set for the A-lines versus B-lines or pleural effusion classification tasks. The set of transformations meeting these criteria were included in AugUS-D, as they were deemed to be essential to performance of downstream classifiers. This methodology is described in detail in Section 3.4 (lines 197-208) and the results are described in Section 4.1 (lines 240-269). Appendix I (lines 746-761) describes the statistical significance testing that was applied to help determine which transformations led to reduced performance when left out from the pipeline.

Comments 5: Provide more detailed guidance or a checklist at the end for developers applying SSL in ultrasound imaging.

Response 5: We thank the reviewer for this comment. We have provided concise guidance for developers applying SSL in ultrasound imaging in the Conclusion of the paper. In response to the reviewer's comment, we have improved the clarity of this guidance and given it an entire paragraph in the conclusion (lines 458-464). 

Reviewer 2 Report

Comments and Suggestions for Authors

This manuscript describes an innovative and sophisticated item, which in simple words could called as "educating AI".

Introduction: fairly the authors introduce the current gaps regarding the reading of radiological images and  proposed and evaluated data augmentation and preprocessing strategies for self-supervised learning in ultrasound.

Results: precise presentation.

Limitations: restricted on lungs disorders

Author Response

Comments 1: Limitations: restricted on lungs disorders

Response 1: Thank you very much for your review. I agree that a limitation of this work is that the experiments were only conducted on tasks within lung ultrasound. We have acknowledged this limitation in the Conclusion (line 459).

Reviewer 3 Report

Comments and Suggestions for Authors

I was glad to read this article, devoted to a very relevant topic of using machine learning in medical diagnostics.

It is worth noting the great work done by the authors, as well as its very detailed description in the manuscript.

Comments and questions.

1. As technical comments, I can note the lack of decoding of some abbreviations: for example, USCL (line 81) and t-SNE (line 274). Perhaps, for the convenience of the reader, it would be better to duplicate the list of abbreviations at the end of the article (due to their abundance).

2. In the captions to figures 4, 6 and 8, there is no division into parts a, b (c), although it is on the figures themselves.

3. As questions on the essence of the article, I would like to clarify why the list of transformations in AugUS-O was limited to the specified 12 points? Have other options been considered or are you considering? For example, is it reasonable to add length/width transformations due to detector features or length/width blurring due to response speed?

4. Do the authors have the opportunity to provide literature data on the efficiency of identifying information in medical ultrasound images (possibly from other areas not related to the article), using machine learning methods and the classical method (i.e. by a doctor). Such information in the introduction can be useful and very illustrative for the reader.

The mentioned comments do not reduce my high opinion of this manuscript, so I believe that it will be useful for a wide range of specialists and can be published in its current form or with minor changes.

Author Response

Comments 1: As technical comments, I can note the lack of decoding of some abbreviations: for example, USCL (line 81) and t-SNE (line 274). Perhaps, for the convenience of the reader, it would be better to duplicate the list of abbreviations at the end of the article (due to their abundance).

Response 1: Thank you for this comment regarding the clarity of the article. To be consistent with the Instructions for Authors in the Bioengineering journal (found at https://www.mdpi.com/journal/bioengineering/instructions), we have ensured that each acronym is defined in the abstract; the main text; and the first figure/table of use. Please see these changes in the following locations: line 82, Table 4 caption, lines 284-285, Figure 5 caption, Figure 8 caption. We also stopped using the acronym "US" for ultrasound, and replaced any instances of this abbreviation with "ultrasound". As an aside, we also added a citation for t-SNE (line 285).

Comments 2: In the captions to figures 4, 6 and 8, there is no division into parts a, b (c), although it is on the figures themselves.

Response 2: We thank the reviewer for this suggestion. Accordingly, we have updated the captions of Figures 4, 6, and 8 to reference the subfigure identifiers.

Comments 3: As questions on the essence of the article, I would like to clarify why the list of transformations in AugUS-O was limited to the specified 12 points? Have other options been considered or are you considering? For example, is it reasonable to add length/width transformations due to detector features or length/width blurring due to response speed?

Response 3: This is an insightful question. AugUS-O was developed by brainstorming a set of transformations that would encapsulate several means by which ultrasound images could vary in in the real world, while minimizing the change that a clinician's impression would be changed after looking at the transformed image. One of the co-authors is an expert in point-of-care ultrasound. We acknowledge that there could be other transformations that could encourage pretrained models to learn useful invariances. To acknowledge this the paper, we have added a sentence that indicates this as a limitation of the work (lines 469-471).

Response 4:  Do the authors have the opportunity to provide literature data on the efficiency of identifying information in medical ultrasound images (possibly from other areas not related to the article), using machine learning methods and the classical method (i.e. by a doctor). Such information in the introduction can be useful and very illustrative for the reader.

Comment 4: We thank the reviewer for raising this interesting and important question. While comparison of physician and AI-based interpretation is not the central aspect of this study, we agree that a small comparison would be helpful to frame this work within the broader context of AI-based ultrasound interpretation. Accordingly, we have added a sentence in the introduction that succinctly states the strong and weak areas of AI-based interpretation and provide a reference to a recent scoping review that discusses the efficacy of multiple AI-based methods in detail (lines 24-27).

Reviewer 4 Report

Comments and Suggestions for Authors

The paper calls: “The Efficacy of Semantics-Preserving Transformations in Self-Supervised Learning for Medical Ultrasound” and concerned of mathematical data processing images of ultrasonic investigation. The article is large array images data with mathematical processing different algorithms.

Questions:

  1. What device authors uses for accruing images?
  2. How resolution effects on data processing?
  3. What software and hardware authors use for mathematical processing?
  4. May be authors can add some unusual fact of imaging processing. How authors choice proper algorithm in image processing?
The advantage of the work is a deep analysis of the problems of incorrect diagnosis of images by mathematical methods using a long list of references (44 items).
Additional question: How does the number of grayscale levels in an image (dynamic range) affect the algorithm's performance and the number of recognition errors?

Author Response

Comments 1: What device authors uses for accruing images?

Response 1: The images in the LUSData dataset were collected retrospectively. Please note that we provide information regarding the images in Table A1. While we do not have information on the probe model number, we are able to provide device manufacturer, probe type, and depth of acquisition.

Comments 2: How resolution effects on data processing?

Response 2: Thank you for this question, as it is important to know how images are preprocessed in machine learning experiments. The images were collected at various resolutions, as they originate from a variety of ultrasound devices, but we standardized the size of the images as a preprocessing step. As outlined in Section 3.5 (line 211), images are resized to 128x128 pixels. This convention was used for prior work for the same problems (A-line versus B-line classification and pleural effusion classification). To make this clear, we have added a clause at line 217 providing a reference to the previous work.

Comments 3: What software and hardware authors use for mathematical processing?

Response 3: Thank you for pointing out that this information is missing. In response, we have provided the Python version used for all code development, provided a link to experimental code, and shared details regarding the hardware used for experiments. Please find this newly added information in Section 3.5 (lines 234-238) and Appendix H (lines 719-722).

Comments 4: May be authors can add some unusual fact of imaging processing. How authors choice proper algorithm in image processing?

Response 4: To isolate the effects of different data augmentation pipelines, we held the architecture constant. Please note that this architecture is consistent with prior work (stated in line 212). For most experiments, we used MobileNetV3Small as the neural network architecture because it can be executed in real-time on mobile devices, which maintains the portability of point-of-care ultrasound. We have added a sentence at lines 213-215 to reinforce this reasoning. Lastly, for completeness, we performed additional experiments using the ResNet18 backbone. These experiments are described in Appendix H. As seen in Table A6, these models overfit when fine-tuned.

Comments 5: How does the number of grayscale levels in an image (dynamic range) affect the algorithm's performance and the number of recognition errors?

Response 5: This is an interesting question for machine learning in ultrasound in general. I imagine that the dynamic range of images would differ by probe type, manufacturer, device model, and device-specific settings. This study is concerned primarily with discovering which data augmentation methods work best for self-supervised pretraining in ultrasound. I believe that the reviewer's question is worthy of its own separate study, as it is an intriguing question that applies to several domains in ultrasound. However, I believe that it is out-of-scope for this investigation.

Round 2

Reviewer 1 Report

Comments and Suggestions for Authors

All revisions are completed.